# Graph Reduction with Unsupervised Learning in Column Generation: A Routing Application

## Abstract

Column Generation (CG) is a popular method dedicated to enhancing computational efficiency in large scale Combinatorial Optimization (CO) problems. It reduces the number of decision variables in a problem by solving a pricing problem. For many CO problems, the pricing problem is an Elementary Shortest Path Problem with Resource Constraints (ESPPRC). Large ESPPRC instances are difficult to solve to near-optimality. Consequently, we use a Graph neural Network (GNN) to reduces the size of the ESPPRC such that it becomes computationally tractable with standard solving techniques. Our GNN is trained by Unsupervised Learning and outputs a distribution for the arcs to be retained in the reduced PP. The reduced PP is solved by a local search that finds columns with large reduced costs and speeds up convergence. We apply our method on a set of Capacitated Vehicle Routing Problems with Time Windows and show significant improvements in convergence compared to simple reduction techniques from the literature. For a fixed computational budget, we improve the objective values by over 9% for larger instances. We also analyze the performance of our CG algorithm and test the generalization of our method to different classes of instances than the training data.

**Keywords**: Unsupervised Learning, Column Generation, Vehicle Routing, Pricing Problem, Graph Reduction, Local Search

## 1 Introduction

Real-life decision-making problems can often be modeled by Combinatorial Optimization (CO). Practical problem instances, however, usually have much larger sizes than instances often encountered in the literature (Luo et al., 2024), while requiring good solutions in limited time. This invokes the need for specialized methods capable of scaling to the large instances of these problems.

Column Generation (CG) is often regarded as an efficient technique to address many CO problems such as Vehicle Routing and Crew Scheduling (Desaulniers et al., 2006), as it keeps the number of decision variables small by iteratively adding variables that contribute most towards the objective value of the problem. To identify variables which contribute to the objective value of a problem, i.e. with non-zero reduced cost, a Pricing Problem (PP) has to be solved. As during the CG algorithm, many Pricing Problems have to be considered, the efficiency of the method lies in the ability to solve the PP efficiently(Václavík et al., 2018). The main challenge in CG is that the PP is often NP-hard, making it difficult to address instances of practical size (Smith-Miles & Lopes, 2012; Hoffman & Ralphs, 2013).

For many CO problems like Vehicle-Routing and Crew-Scheduling, the PP is often an Elementary Shortest Path Problem with Resource Constraints (ESPPRC) (Morabit et al., 2023). This is one of many PPs that can be modeled as a graph. Several recent works have studied the use of Machine Learning (ML) techniques to solve large-scale CO problems with CG such as in Morabit et al. (2021), Morabit et al. (2023) and Xu et al. (2023). These methods have a good performance on instances up to 400 customers, but may struggle to scale on larger instances nearing 1000 customers. Scaling to larger instances, indeed, still remains a challenge for many ML applications in CO (Bogyrbayeva et al., 2022). Not to mention the computational

burden associated with training such large models. To that end, an efficient technique is needed by which the pricing problem can be efficiently solved for larger instances. Graph Reduction (GR) tools are a good nominee as they reduce the problem size by simply eliminating arcs that are unlikely to be in the optimal solution, helping to preserve computational efficiency.

The idea is to retain the most promising arcs, likely to be the optimal solution, while discarding the rest. In theory, it is not easy to establish general mathematical rules to decide whether a node or an arc should be included/excluded. One simple **heuristic** rule is to eliminate all nodes with zero dual values as they do not improve the reduced cost while consuming resources (Barnhart et al., 1998). However, the number of such nodes is small for many PP iterations and stronger reduction methods are needed.Alternative simple rule-based techniques may be applied, such as the GR heuristics in Xu et al. (2023). While these heuristics generally improve the speed of solving the PP, they often eliminate important arcs, affecting the quality of the generated columns and the convergence speed. They also disregard graph connectivity, preventing the generation of feasible columns.

In turn, we seek to design a reduction method that sufficiently reduces the PP instances while retaining the most promising arcs such that columns with large non-zero reduced costs can be generated. In this paper, we propose a framework for GR methods that aims to balance this trade-off. Specifically, we make use of a ML model for reducing the graph of the pricing problem in CG to solve large-scale problems. Our model generalizes the model in Min et al. (2024) to a more complex setting. To illustrate the strength of the proposed method, we compare it with other reduction techniques on instances of the Capacitated Vehicle Routing Problem with Time Windows (C-VRPTW).

Our paper is organized as follows. Section 2 discusses previous work for solving the PP, alongside GR techniques. Section 3 describes the problem we address. Section 4 presents our framework and the ML model for GR in greater detail. Section 5 presents our case study and associated numerical experiments and results. Section 6 asserts our conclusions.

## 2  Previous Work

CG is a well-known optimization method that can efficiently solve many CO problems of moderate sizes. There are various exact and approximate methods that could be used to solve the pricing problems appearing in CG for moderately sized instances, such as Dynamic Programming (DP)-based methods and simple local search heuristics.

DP-based methods, like labeling algorithms (Desrochers et al., 1992; Feillet et al., 2004; Boland et al., 2006) are the most popular in the literature for solving PPs. They approximate the solution space by relaxing some constraints and introducing dominance criteria to find dominant columns. These approximations, however, may lead to weak lower bounds as explained in Feillet et al. (2004). Exact variants of labeling algorithms or DP-based methods as the ones proposed in Feillet et al. (2004) and Lozano et al. (2016) may strengthen the bounds, however, these methods can not easily cope with the exponentially increasing search space often associated with much larger instances. Furthermore, they require complex fine tuning in order to perform efficiently.

Local search heuristics, like the Tabu Search used in Dabia et al. (2017) and Lozano et al. (2016) or the heuristic described in Guerriero et al. (2019), can handle relatively larger instances as they do not iterate over the possible ordering of nodes in a solution. However, they may often lead to infeasible columns or columns with small reduced-costs being generated and therefore are mostly used to accelerate DP-based methods.

ML models have the capacity to process complex graph information for decision-making in a principled manner (Bengio et al., 2021). Hence, we resort to ML-based GR techniques that aim to reduce an instance size without significantly jeopardizing the quality of the obtained solutions.

Morabit et al. (2021) was among the first to lay a foundation for CG and ML. They propose a model that uses labeling algorithms to solve the PP and build a training dataset for a ML algorithm that selects the columns generated to be added to the Master Problem (see Section 3) for a VRP. While the ML architecture used

manages to reduce the computational burden compared to CG, it is rather expensive to train. Furthermore, as it relies on a labeling algorithm, the generation of the training data and of the solutions can be time consuming.

ML has previously been applied for GR, for example in Morabit et al. (2023), Xu et al. (2023) and Min et al. (2024). In the first paper, a Graph Neural Network (GNN) employs a Supervised Learning (SL) module to learn which arcs should be retained in the reduced PP. In the second paper, the GNN uses Reinforcement Learning (RL) to decide on the reduction heuristic to be applied from a set of heuristics. Both methods displayed notable computational savings, although they entailed long training times and suffered inconsistent performance, especially among larger instances.

The method of Min et al. (2024), which we refer to as SAG, is shown to be the most promising. The authors propose a GNN that identifies the arcs most likely to be in the optimal solution of a Traveling Salesman Problem (TSP) and employ a local search to construct the final tour. On instances of up to 1000 customers, the number of arcs was on average reduced by around 90% with 99% of "optimal" arcs retained in the solution. SAG leverages an attention-mechanism architecture which has repeatedly delivered impressive results in studies like Kool et al. (2018), but is computationally demanding to train for RL and SL. With Unsupervised Learning (UL), SAG uses 10% of the parameters and 0.2% of the training samples compared to SL and RL (Min et al., 2024).

In this paper, we propose an UL model based on SAG for reducing the pricing problem in column generation. In particular, we extend the SAG model to be able to solve more complex problems than TSP. More precisely, we generalize the heat-map generation method to cope with a PP solution structure where not all customers are visited, while incorporating other constraints that are not present in TSP. Additionally, we devise a new local search algorithm that speeds up convergence with a minimal number of iterations. We showcase how our proposed method can be used to reduce and solve PPs in CG for routing - without loss of generality - thereby bridging a new framework between CG and GR.

## 3 Problem Description

For simplicity, we assume we are dealing with a minimization problem, although our framework is also applicable to maximization problems. The main idea of CG is to decouple a linear program (LP) into two problems, a Restricted Master Problem (RMP) and a Pricing Problem (PP). The RMP considers only a subset of the decision variables to keep the number of decision variables small and reduce the associated computational burden. This subset is determined by repeatedly solving the PP in consecutive CG iterations. At a given iteration, the PP retains information from the RMP that can be used to identify variables with potential to reduce the RMP's objective value. The goal of the PP is to determine variables/ columns with a negative reduced cost. The objective value of the PP corresponds to the reduced cost of the column it generates in the corresponding iteration. Lower negative reduced costs imply faster convergence (Lübbecke & Desrosiers, 2005). Once a column with negative reduced costs is found by the PP, it is added to the RMP which is then re-solved to provide the PP with new information in the next iteration. The procedure continues until no more variables with negative reduced costs can be found or when some termination criteria - like the maximum number of iterations - is reached. For more details on column generation we refer to Feillet (2010).

In this work, we restrict our attention to graph-based PPs like ESPPRC. The objective of ESPPRC is to find the shortest feasible path from an origin node to the destination node, where feasibility is dictated by the resource constraints and the requirement that all nodes are visited at most once. A complete formulation for ESPPRC can be found in Feillet et al. (2004). ESPPRC is NP-Hard (Dror, 1994), making it computationally intractable for large instances.

In this paper, we approximately solve the PP by, first, reducing the number of arcs considered in the problem before deploying a simple heuristic to treat the reduced version. Our objective is to reduce the PP to such an extent by which it can be efficiently solved by a heuristic without losing the arcs that are most able to reduce the objective value. In doing so, we account for arc feasibility and graph-connectivity. This is in contrast

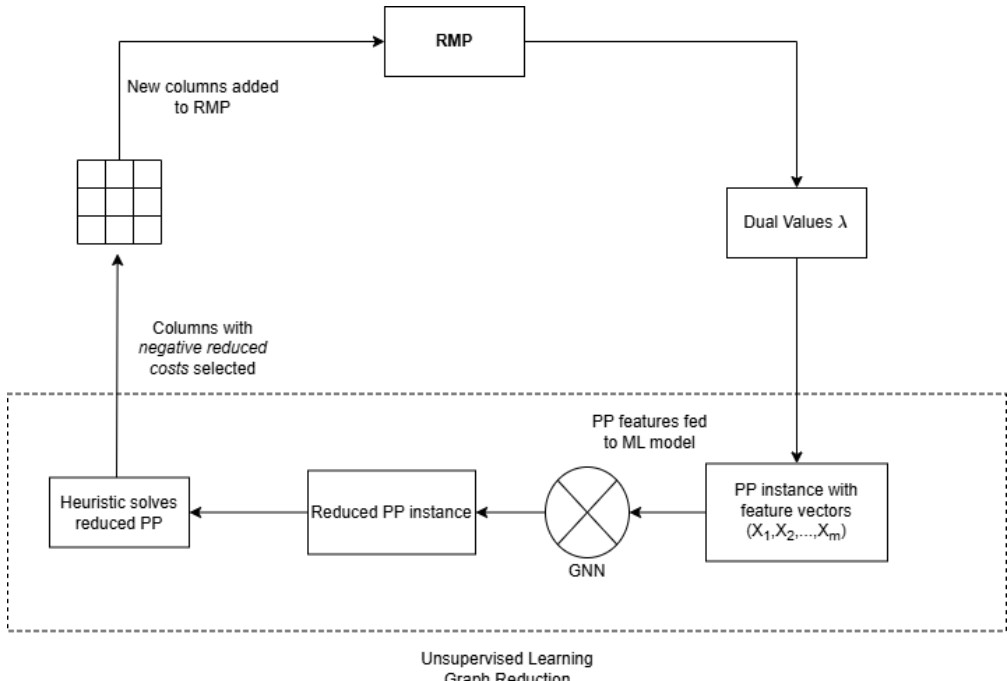

**Figure 1** Illustration of the proposed framework with a pre-trained GNN.

to previous works such as Desaulniers et al. (2008), Dell'Amico et al. (2006) and Santini et al. (2018) where simple reduction heuristics have been applied to the PP, disregarding arc feasibility and connectivity.

In contrast to supervised learning, our ML method strives to select the arcs in an unsupervised manner by estimating their likelihood of belonging to the optimal solution without knowing the optimal solution. This is done by simply processing information on the PP instance and selecting the best arcs based on criteria established during training. Figure 1 illustrates our proposed framework, given a pre-trained GNN. The figure simply explains how our UL model is configured to the CG loop described at the beginning of this section. In a given CG iteration, the duals produced by the master problem are used alongside the other problem parameters as input features to our GNN. The GNN then decides on the most promising arcs to be retained by the PP which is solved by a simple heuristic to produce a set of columns. These columns are added to the master problem and a new iteration begins. In the following section, we explain the rationale underlying our graph reduction mechanism.

## 4 Heat-Map-Guided Local-Search for Column Generation

Let $\mathcal{V}$ and $\mathcal{A}$ be the set of nodes $i$ and arcs $(i,j)\forall i,j \in \mathcal{V}$ for a given PP instance. Consider the probability that an arc $(i,j) \in \mathcal{A}$ belongs to the optimal solution of the PP. For TSP, Min et al. (2024) estimate this probability by means of a ML model that maps graph features into a probability distribution for arcs. This distribution is then used to construct a heat map that accounts for graph connectivity, and is then coupled with a local search. The heat map guides the local search by assigning arcs higher probabilities based on their length and connectivity to other arcs with high probabilities from the initial distribution in $\mathcal{T}$ produced by the ML model. Once constructed, the heat-map is further adjusted to reduce the number of arcs considered by the local search.

However, TSP is quite distinct from ESPPRC which is more complex as it is characterized by more constraints. An important feature of ESPPRC, in general, is that often not all nodes must be visited in a solution, unlike in TSP. This not only requires changes to the generation of the heat map, but also to the local search design as standard procedures for VRP and TSP such as 'k-opt' algorithms (Bräysy & Gendreau, 2005) may not work anymore due to their assumption that all nodes are visited in a solution.

In the following sections, we explain how the heat map in Min et al. (2024) is generated and the generalization we propose in order to solve PPs in CG. The proposed generalization mechanism can be extended to other problems that can be modeled as graphs as well.

## 4.1 Heat Map Generation

The SAG model proposed in Min et al. (2024) takes as input graph features related to both nodes and arcs, and produces an estimated probability matrix $\mathcal{T}$ of dimensions $n \times n$ with $n = |\mathcal{V}|$, the number of nodes in the graph. Element $(i, j)$ in $\mathcal{T}$ corresponds to the probability that arc $(i, j)$ belongs to the optimal solution. This probability, however, considers arc $(i, j)$ individually without accounting for connectivity to other arcs. Ideally, as the solution to the PP must be connected, graph connectivity should be retained as much as possible. Therefore, the heat map $\mathcal{H}$ proposed by the authors, takes arc connectivity into account by the following definition:

$$\mathcal{H} = \sum_{t=1}^{n-1} h_t h_{t+1}^T + h_n h_1^T \tag{1}$$

where $h_t$ is the $t^{th}$ column of $\mathcal{T}$ and $h_t^T$ is its transpose. While $\mathcal{H}$ - which also has dimensions $n \times n$ - may result in a better-connected graph than $\mathcal{T}$, this is not strictly enforced. Consequently, Min et al. (2024) resort to a local search procedure that aims to interchange arcs with more promising ones to gradually improve the solution's objective value. We discuss the adjustments to the method by which the heat map is constructed and the local search for PPs in further detail in the following section.

## 4.2 Extension to Pricing Problems

We leverage the same training algorithm as in SAG, which requires a set of predefined training instances on which the model is trained in every epoch. Specifically, SAG uses a fixed set of 3,000 TSP instances for training. Since TSP is a relatively simple problem, SAG is able to explicitly model the objective function and the associated constraints in one loss function where constraint violations are penalized.

To generate the heat map for a PP, we configure the constraints in the loss function **implicitly**. Specifically, we consider three factors related to each arc, its **(1) feasibility**, **(2) contribution to the objective value** and **(3) contribution to constraints**. A good reduction method should strive to keep only feasible arcs. Some arcs may be infeasible due to operational constraints such as time-windows, or due to pruning strategies invoked by the branch-and-bound procedure as mentioned in Feillet (2010). Further, each feasible arc is characterized by length $v_{ij}$ and a contribution $r_{ij}^k$ to constraint $k \in K$, where $K$ is the set of constraints. $v_{ij}$ and $r_{ij}^k$ represent the parameters corresponding to arc $(i, j)$ in the objective function and in constraint $k$ in the mathematical formulation of the PP. The contribution of arc $(i, j)$ to the objective value is determined by the arc lengths $v_{ij}$, while the contribution to constraint $k \in K$ is determined by $r_{ij}^k$.

An important advantage of our method is that it balances the added value of the selected arcs in reducing the objective value with the degree by which the associated arc constrains the search space. For instance, we may be inclined to pick an arc that has a smaller contribution to reducing the objective value if it does not constrain the solution heavily with its inclusion, as it may allow for a larger search space.

### 4.2.1 Loss Function

Based on the previous discussion, we modify the loss function in Min et al. (2024) according to the procedure described below. We first pre-process the instances in our training data as follows. For each infeasible arc $(i, j)$, we set the contribution to the objective value to some high penalty term.The resulting weight in the training loss is given by $q(v_{ij}, r_{ij})$ where $q(.)$ is some discount function of the arc length $v_{ij}$ and vector $r_{ij}$ whose elements are constraint contributions $r_{ij}^k$, $k \in K$. The function $q(.)$ is assumed to be decreasing in all arguments. The output of $q(.)$ is stored in a square matrix $Q$ of length $n$ with elements $q_{ij}$ corresponding to arc $(i, j)$.

The loss function is now defined as follows:

$$\mathcal{L}(\Theta) = \lambda_1 \sum_{(i,j) \in \mathcal{A}} \mathcal{H}_{ij} q_{ij} + \lambda_2 \sum_{i \in \mathcal{V}} \mathcal{H}_{ii} + \lambda_3 \sum_{i \in \mathcal{V}} (1 - \sum_{j \in \mathcal{V}} \mathcal{T}_{ij})^2 \tag{2}$$

with $\Theta$ being the set of parameters of the GNN and $\lambda_1$, $\lambda_2$ and $\lambda_3$ are positive penalty terms. The first sum encourages the selection of (feasible) arcs with the smallest length and constraint contribution. The second sum penalizes the selection of diagonal elements in $\mathcal{H}$ as all nodes can be visited at most once. The last sum encourages a column-wise normalization for the probabilities of incoming arcs in $\mathcal{T}$. This is analogous to the row-wise normalization of the probabilities of outgoing arcs imposed by the softmax function in the final layer of the SAG network. In summary, the loss function strives to balance the row-wise and column-wise normalized sum through the $q_{ij}$ weights while disregarding infeasible arcs as much as possible.

### 4.2.2 GNN Input

The input features fed to SAG to generate $\mathcal{T}$ and, accordingly, $\mathcal{H}$ are constituted of node and arc features of the graph instance. Let $N_e$ and $A_e$ define the set of node features and arc features in the graph used by our model to decide which arcs to keep in the graph. The components of $N_e$ and $A_e$ largely depend on the problem at hand, since PPs can vary in structure.

For ESPPRC, an important node feature is the dual value associated with node as these guide the CG procedure (Feillet, 2010). These are basically decision variables from the dual problem which do not have a straightforward distribution that can be used to sample training data. Instead, they are generated during the CG procedure, unlike the other problem parameters which are available during training. The presence of dual values poses a considerable difference to TSP where all problem parameters are available during training.

To generate a representative set of the dual values appearing in each CG iteration, we use the following method from Abouelrous et al. (2025). Here, the dual values are sampled from the interval $[0, \theta \times t_i^{max}]$ where $t_i^{max}$ is the maximum travel time to node $i$ and $\theta$ is a scaling factor sampled uniformly from the interval $[0.2, 1.1]$. The lengths of the intervals from which the duals are sampled are varied to capture the variability in duals that is often present in different CG iterations.

Other examples of node features $\in N_e$ in ESPPRC include the x,y coordinates in 2-D space, customer demands and time windows. For arc features $A_e$, we use the $Q$ matrix specified in Section 4.2.1. Using $Q$ for $A_e$ poses another distinction to TSP as the more complex constraints of ESPPRC are implicitly configured in the arc features as well, enabling SAG to determine the most rewarding arcs with minimal resource consumption from the problem input.

### 4.2.3 Heat Map Adjustment

We retain the top $M$ elements in each row of $\mathcal{H}$ and set the rest to zero to retain a reduced heat map $\overline{\mathcal{H}}$ - as done in Min et al. (2024). Since it is a strict requirement for each path in ESPPRC to start at an origin node and end at a destination node, we retain all outgoing arcs from the origin node and all incoming arcs to the destination node. This guarantees that there is always a path from the origin to the destination.

Afterwards, we symmetrize $\overline{\mathcal{H}}$ to generate $\mathcal{H}'$ as follows:

$$H' = \overline{\mathcal{H}} + \overline{\mathcal{H}}^T \tag{3}$$

with $\overline{\mathcal{H}}^T$ being the transpose of $\overline{\mathcal{H}}$. We normalize the rows of $\mathcal{H}'$ to sum to 1 so that they form a probability distribution from which we can sample neighboring nodes $j \in \mathcal{V}$ given the node in row $i$.

### 4.2.4 Local Search

We couple the adjusted heat map described in Section 4.2.3 to a local search algorithm similar to that of Guerriero et al. (2019). The local search is characterized by a series of exchange operations where each

operation involves either the addition of a arc to the path, the removal of an arc or the interchange of two arcs.

The local search starts from a feasible solution generated by a construction heuristic. The local search chooses randomly a node $u$ from the current path and then chooses randomly a neighboring node $v$ as per $\mathcal{H}'$. If $v$ is the destination node, it is added right after $u$ and the new path ends at $v$. If $v$ is not on the path, a new path is formed by inserting $v$ between $u$ and the succeeding node $o$ in the original path. If $v$ is on the path and $v$ is a successor of $o$, a new path is formed by removing $o$ in the original path. Otherwise, the new path is formed by interchanging $v$ and $o$.

The path with the lowest reduced costs among the set of new feasible paths is selected as the current path. The procedure is repeated for a fixed number $C_e$ of iterations before the final current path is selected. To speed up convergence, we run the procedure on multiple threads where each thread starts with a different initial path and optimizes it through the aforementioned local search. As a result, multiple columns are generated of which the ones with negative reduced costs are added to the RMP. This parallelization mechanism is similar to the one in Lozano et al. (2016). The training of SAG is summarized in Algorithm 11, while testing - with a pretrained SAG model - is summarized in Algorithm 2. For simplicity, we refer to the overall procedure described in Algorithm 2 as Unsupervised Learning Graph Reduction (ULGR).

---

**Algorithm 1** SAG Training Procedure

---

1: **Input:** Training dataset $\Omega$, batch size $B$, weights $\lambda_1$, $\lambda_2$, $\lambda_3$ and nr. of training epochs $E$
2: **Output:** Trained SAG model
3: **Initialization:** Initial GNN parameters
4: Pre-process instances as described in Section 4.2.1.
5: **for** epoch $e \in E$ **do**
6:  **for** batch of size $B$ from $\Omega$ **do**
7:   Provide input features as specified in Sections 4.2.2 to SAG.
8:   Calculate loss function from (2) using weights $\lambda_1$, $\lambda_2$, $\lambda_3$ and $\Omega$.
9:   Back-propagate loss.
10:  **end for**
11: **end for**

---

**Algorithm 2** Heat-Map Guided Local Search for Column Generation

---

1: **Input:** Predefined testing dataset $\overline{\Omega}$, pretrained SAG model and Nr. of local search iterations $C_e$.
2: **Output:** Solutions $\mathcal{S}$ for problems $\overline{\omega} \in \overline{\Omega}$.
3: **for** $\overline{\omega} \in \overline{\Omega}$ **do**
4:  **while** CG scheme generates variables with negative reduced costs **do**
5:   Generate $\mathcal{H}$ from pretrained SAG model.
6:   $\mathcal{H}' = Heat\_Map\_Adjustment(\mathcal{H})$ from Section 4.2.3.
7:   Generate initial solution to PP using construction heuristic.
8:   Use $\mathcal{H}'$ in combination with local search from Section 4.2.4 to solve PP.
9:   **if** the resulting variable from solving PP has a negative reduced cost. **then**
10:    Add variable to master problem
11:   **else**
12:    Terminate CG Scheme and return current solution to RMP $s$.
13:   **end if**
14:  **end while**
15:  $\mathcal{S} = \mathcal{S} \cup \{s\}$
16: **end for**
17: **return** $\mathcal{S}$.

---

# 5 Numerical Experiments

In our experiments, we solve a VRP variant with capacity constraints and time windows - also known as C-VRPTW - using a CG scheme. In this case, the PP is a variant of ESPPRC with time windows, of which a complete formulation is given in Chabrier (2006). In the following sections, we present the set-up of our numerical experiments and analyze the performance on different datasets.

## 5.1 Data Generation

To illustrate the ability of our method to preserve feasibility, we generate highly constrained C-VRPTW instances as follows. We sample the node coordinates uniformly at random from a square of length 1. Customer demands are uniformly sampled as integers from the interval [1,10]. For instance sizes 200 and 500, we consider a vehicle capacity of 50 and for size 1,000 a capacity of 80. Service times are sampled uniformly from $[0.2, 0.5]$. Travel times are assumed equal to the euclidean distances between nodes. Time-windows $[a, b]$ are sampled such that each time-window has a length of either 1 or 2 time units with $a$ sampled uniformly as integer from the interval [0,16]. The time-window of the depot is [0,18]. The dual values are generated according to the procedure in Section 4.2.2.

All input parameters of the problem are scaled as follows. The travel times, service times and time windows are scaled by the depot's upper time window and the customer demands are scaled by the vehicle capacity. Furthermore, in the PP each arc is characterized by a length $p_{ij} = t_{ij} - d_j$ with $t_{ij}$ being the travel time from node $i$ to $j$ and $d_j$ the dual value associated with node $j$. The $p_{ij}$ values are scaled by $\max\left(|\min_{(i,j)\in\mathcal{A}}(p_{ij})|, |\max_{(i,j)\in\mathcal{A}}(p_{ij})|\right)$ so that they all fall in the interval [-1,1]. The SAG model takes the scaled customer coordinates, demands, service times, time windows and dual values as input for $N_e$ and the $p_{ij}$ values as input for $A_e$.

To define the loss function, we set the function $q_{ij}(.)$ in (2) to $p_{ij}e^{(-u_{ij}-d_j)}$ if $p_{ij} < 0$ where $u_{ij}$ is the minimum time consumption needed to traverse arc $(i, j)$ taking into account the time windows of nodes $i$ and $j$, the service time of node $i$ and the travel time from node $i$ to $j$ and $d_j$ is the demand of node $j$. Both $u_{ij}$ and $d_j$ are scaled by the depot's upper time window $b_0$ and the vehicle capacity $Q$. If $p_{ij} > 0$, we set $q_{ij} = p_{ij}e^{(u_{ij}+d_j)}$. For infeasible arcs - as per time window restrictions, $q_{ij} = 2$. This represents a relatively large penalty term, as all other feasible arcs are scaled to be in the range [-1,1].

## 5.2 Baseline

We compare our method to the reduction heuristic mentioned in Santini et al. (2018). We chose this technique due to its simplicity as well as its superior performance compared to other reduction heuristics, as illustrated in (Xu et al., 2023) where it is referred to as BE2. The technique retains the $\beta$ percent of arcs with lowest value, or maximum contribution to objective value, while discarding the rest. We set $\beta$ to 0.2. This number aligns with the value used in the experiments of Santini et al. (2018). Note that in this method, the feasibility of the arcs and their contribution to the constraints is disregarded.

As ESPPRC are commonly solved by DP- based heuristics (Desrochers & Soumis, 1988), we chose a DP-based heuristic as baseline for solving the PP after reduction. Starting from the depot, the heuristic initiates a parallel thread for each customer and explores outgoing arcs through their sorted lengths - as in Lozano & Medaglia (2013) - such that arcs with smallest length are visited first. To speed up the PP, we used the rollback pruning strategy in Lozano et al. (2016) and rejected all routes that had a significantly positive reduced cost after 75% of any resource was consumed.

Each thread terminates as soon as it either finds a column of negative reduced costs less than a certain limit $P_{lb}$ - which we set to -1 - or after a certain time limit - which we set to 30 seconds - has elapsed without finding such columns. Furthermore, the algorithm also terminates if 20 threads manage to find paths with length $\leq P_{lb}$ or if 100 threads terminate due to the above conditions. This prevents unnecessarily long runtimes with larger numbers of threads. The heuristic does not accept any columns with non-negative reduced costs. We found this DP heuristic to be much faster than standard labeling algorithms which struggle with larger instance sizes due to the generation of many labels (Engineer et al., 2008).

For the construction heuristic used to initialize a solution before the local search, we use the same DP-based heuristic. The difference is that we set a time limit of 5 seconds and a $P_{lb}$ limit of -0.1. Furthermore, columns with reduced costs up to 0.5 are accepted. The rationale behind this is that, in the presence of a smaller time-limit and target lower bound $P_{lb}$, our heat map can significantly improve the reduced costs with a small number $C_e$ of exchange operations, even when the initial columns may have positive reduced costs - up to 0.5.

For our method and the baseline, we make use of a compute capacity of 100 simultaneous threads and set the overall time limit for solving the root node of the C-VRPTW instance to 1 hour. We also use the same CG initialization procedure for both methods. We start with a set of columns representing a feasible solution constructed by a greedy mechanism where the nearest unvisited customer is added to the current route until all customers are visited.

## 5.3 Results

We consider three classes of C-VRPTW instances of sizes 200, 500 and 1000. We train three different models for each class. The instance sizes are in line with the ones in Min et al. (2024). Training was conducted on a GPU node with 2 Intel Xeon Platinum 8360Y (Intel, 2025) Processors and a NVIDIA A100 Accelerator (Nvidia, 2025).

For the instance sizes 200, 500, and 1000, the training times were approximately 16, 38 and 140 **minutes**, respectively. SAG requires very little computational resources for training. In principle, training can be performed with a small number of GPUs. The most influential factors in training time are the data size and the batch size. We set the size of the training dataset to 5,000 instances, while the batch size varied between 32 and 64. These values are in line with (Min et al., 2024). The sample-efficiency of SAG during training is noteworthy. Unlike other attention-based models in Kool et al. (2018) and Kwon et al. (2020), SAG does not require a lot of training samples. Furthermore, the number of parameters in the UL module is significantly less, leading to much shorter training times. This also enables us to train models for larger problem instances than observed in these works.

The experiments with the CG scheme were carried out on an AMD EPYC 9654 (AMD, 2025) cluster CPU node. We consider a limited number of $C_e = 20$ exchange operations for our local search. The computational gain associated with such a small number of exchange operations can be largely attributed to the heat map's ability to retain the most rewarding arcs.

For testing the methods, we consider $R = 50$ test C-VRPTW instances for each size $n \in \{200, 500, 1000\}$. For each node, we select the top $M = 10$ arcs to generate the adjusted heat map in accordance with the procedure in Section 4.2.3. We consider the following metrics for assessment: $obj_{Gap}$ and $t_{Speed-up}$. The former refers to the average relative difference between the $obj_{ULGR}$, the final objective value of the C-VRPTW root node obtained by our UL model, and $obj_{BE2}$, the objective value attained by the BE2 technique. It is calculated as follows:

$$obj_{Gap} = \frac{1}{R} \sum_{r=1}^{R} \frac{(obj_{ULGR}^r - obj_{BE2}^r)}{obj_{BE2}^r}, \tag{4}$$

where the superscript $r$ indicates the $r^{th}$ instance. The second measure, $t^{Speed-up}$, refers to the average ratio between the time ULGR needed to reach the final objective value of the BE2 technique, $t_{ULGR}^r$, and the time it took BE2 $t_{BE2}^r$ for instance $r$ and is calculated by:

$$t_{Speed-up}^{obj_{BE2}<obj_{ULGR}} = \frac{1}{R} \sum_{r=1}^{R} \frac{t_{BE2}^r}{t_{ULGR}^r}. \tag{5}$$

Since the baseline BE2 results in a higher objective value for some instances, the metric in (5) is undefined. In such cases, we consider the time it takes ULGR to reach the final objective value of BE2, whose average ratio to BE2's total run-time is given by the quantity $t_{Speed-up}^{obj_{ULGR}<obj_{BE2}}$. We also report the fraction of instances where this happens $J(<)$. The results are given in Table 1.

| $n$ | $\mathbf{obj_{Gap}}$ | $J(<)$ | $\mathbf{t_{Speed\text{-}up}}^{obj_{ULGR}<obj_{BE2}}$ | $\mathbf{t_{Speed\text{-}up}}^{obj_{BE2}<obj_{ULGR}}$ |
|---|---|---|---|---|
| 200 | -6.10% | 45/50 | 0.52 | 0.94 |
| 500 | -2.05% | 45/50 | 0.74 | 1.00 |
| 1000 | -9.20% | 50/50 | 0.73 | – |

**Table 1** Results of ULGR compared to the DP baseline with BE2 reduction technique on simulated data.

For $n = 200$, ULGR improves the objective value by 6.10% on average. ULGR outperforms the baseline in 45 of the 50 instances. For those instances, ULGR takes on average 52% of the baseline's total run-time to reach its objective values. For the other 5 instances, the baseline's run-time to reach ULGR's final objective values is similar to ULGR's run-time, standing at 94% of ULGR's total run-time.

For $n = 500$, ULGR improves the objective value by a smaller margin of 2.05%. This percentage is less compared to $n = 200$ likely due to the increased difficulty of solving larger instances. This is also indicated by the fact that ULGR takes around 76% of the baseline's run-time to reach its objective values for the 45 instances where it has a lower objective, while for the other 5 instances, the run-times are pretty similar as demonstrated by the ratio of 1.

In the most challenging case, corresponding to $n = 1000$, ULGR significantly improves the objective value by 9.20% on average and outperforms the baseline in all instances, taking 73% of its run-time to reach its objective values. The divergent objective gap could be attributed to the baseline's failure to efficiently scale to significantly larger instances.

It is worth mentioning that we could not provide an optimality gap as most of the problems we deal with can not be solved to optimality in any reasonable run-time. This also holds if one tried to generate the set of all feasible routes (disregarding CG) and solve the associated LP-relaxation. Generating this set alone is extremely time-consuming due to the large number of feasible routes. Rather, we focus our analysis on the best bound that can be generated within a practical computational budget.

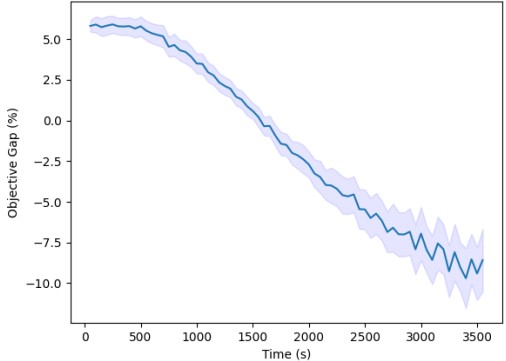

**(a)** Convergence plot of $obj_{Gap}$ over time with relative to BE2 confidence intervals for $n = 1000$.

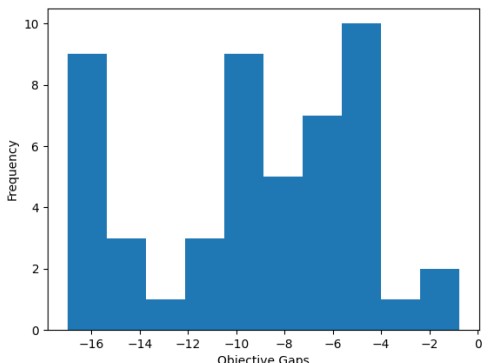

**(b)** Distribution of the values of $obj_{Gap}^r$ relative to BE2 over all $K = 50$ instances.

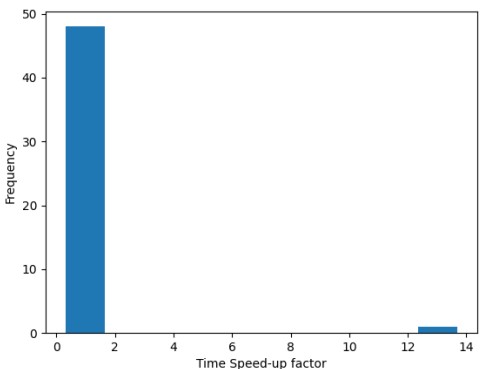

**(c)** Distribution of the values of $t_{Speed-up}^{r,obj_{ULGR}<obj_{BE2}}$ over all $K = 50$ instances.

**Figure 2** Comparison of ULGR to BE2

The BE2 strategy retains by default 20% of arcs in a graph specified by the $\beta$ parameter we chose. For instances of size 200, 500 and 1,000 this is equal to roughly 8000, 50,000 and 200,000 arcs. On the other hand, the UL model retains a number of arcs equal to $(M+2)n =$ 2,400, 6,000 and 12,000 arcs with $M = 10$ for the given values of $n$, where the extra 2 is due to the retention of all the depot's arcs. These numbers are significantly less than the ones resultant to BE2.

Setting the BE2's $\beta$ parameter to give a similar number of arcs as ULGR results in an over-reduction of the graph whereby only a few routes with negative reduced costs can be generated. This is because the BE2 strategy does not take into account arc feasibility which can sometimes lead to the retention of infeasible arcs, elimination of important connecting arcs or even graph disconnectivity. Even for the given value of $\beta = 0.2$, premature convergence was still observed in some instances due to these matters. Higher values of $\beta$, on the other hand, resulted in negligible reduction in both graph size and run-time. Our experiments also demonstrated that the value 0.2 led to a reasonable balance between the aforementioned phenomena.

Figures 2a, 2b and 2c plot the convergence of $obj_{Gap}$ along time, its distribution as well as the distribution of $t_{Speed-up}^{obj_{ULGR}<obj_{BE2}}$ for $n =$ 1000 against BE2. In Figure 2a, the light blue shades represent confidence intervals. We observe that the decline in objective value - and thus $obj_{Gap}$ is more or less constant through time, but increasingly variable as shown by the wider confidence interval due to BE2's failure to scale and early termination. Meanwhile, the large objective improvement of 9.20% is due to ULGR consistently outperforming BE2 among all instances rather than outliers.

On the other hand, the realized $t_{Speed-up}^{obj_{ULGR}<obj_{BE2}}$ ratio at 0.73 is subject to an outlier, which when discarded, brings the ratio down to 0.46. This indicates the dramatically faster convergence of ULGR for $n = 1000$. The results displayed a consistent pattern where BE2 converges rapidly at the beginning before slowly stalling while ULGR maintains a fixed convergence rate. The realized outlier is due to a fast start for BE2 that is met with premature termination later due to the infeasibility issues posed above.

For reference, we carry out the same comparison with the same DP baseline when no GR is involved. The respective objective value and run-time for this baseline are referred to as $obj_{NoGR}^r$ and $t_{NoGR}^r$ for instance $r$. The results are given in Table 2.

| $n$ | $\mathbf{obj_{Gap}}$ | $J(<)$ | $\mathbf{t_{Speed-up}}^{obj_{ULGR}<obj_{NoGR}}$ | $\mathbf{t_{Speed-up}}^{obj_{NoGR}<obj_{ULGR}}$ |
|---|---|---|---|---|
| 200 | -3.79% | 50/50 | 0.49 | – |
| 500 | -2.11% | 49/50 | 0.76 | 1.02 |
| 1000 | -2.45% | 45/50 | 0.78 | 0.90 |

**Table 2** Results of our GR technique compared to the DP baseline with no reduction on simulated data.

For $n = 200$, compared to the case where no reduction is performed, ULGR improves the objective value in all 50 instances, with an average objective decrease of 3.79%, while assuming only 49% of the run-time. For $n = 500$, the average objective gap is at 2.11% while ULGR assumes 76% of the run-time to reach similar objective values. Lastly, for $n = 1000$, ULGR is better in 45 of the 50 instances with an average objective gap of almost 2.50%, while assuming 78% of the run-time. For the other 5 instances, the baseline without reduction method is slightly faster taking 90% of our run-time.

Analogous to the analysis above, Figures 3a, 3b and 3c provide the same analysis without GR for the case $n = 1000$. Similar to the comparison with BE2, our method shows a consistent convergence rate (see Figure 3a) compared to the case where no reduction was performed. The difference is in the confidence intervals which are wider in earlier iterations and narrower thereafter. This is due to the slow start associated with the method without reduction which gains speed in later iterations and its performance stabilizes.

For the distributions of $obj_{Gap}^r$ and $t_{Speed-up}^{r,obj_{ULGR}<obj_{NoGR}}$, we see that the results are due to ULGR consistently obtaining better objective values in shorter computation times, where the values of $t_{Speed-up}^{k,obj_{ULGR}<obj_{NoGR}}$ are evenly dispersed around the average of 0.78.

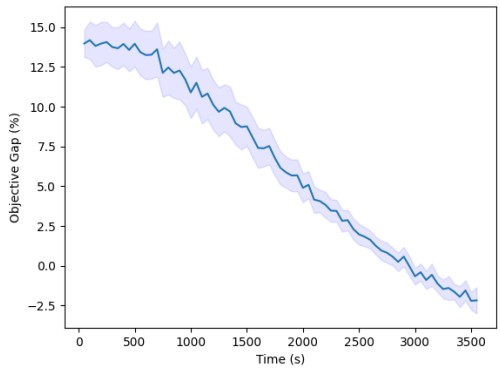

**(a)** Convergence plot of $obj_{Gap}$ over time with relative to no reduction confidence intervals for $n = 1000$.

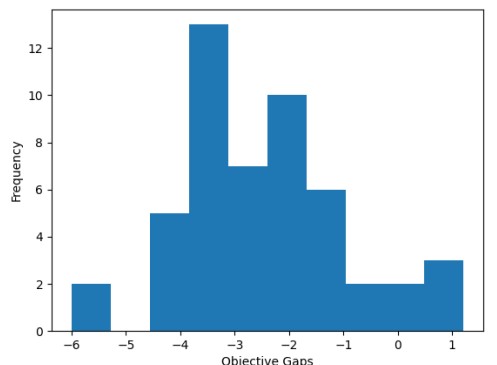

**(b)** Distribution of the values of $obj^r_{Gap}$ relative to no reduction over all $K = 50$ instances.

**(c)** Distribution of the values of $t^{r, obj_{ULGR} < obj_{NoGR}}_{Speed-up}$ over all $K = 50$ instances.

**Figure 3** Comparison of ULGR to no reduction.

The performance disparity between the BE2 reduction method and the method without reduction is obvious. For $n = 200$ and $500$, although BE2 outperforms ULGR in more instances, it struggles to scale for $n = 1000$ where the method without reduction even manages to outperform ULGR in a few instances. The large objective gaps obtained by BE2 for $n = 200$ and $1000$ are due to premature termination as explained above.

ULGR strives to retain the arcs giving better objective values without jeopardizing solution feasibility as opposed to the simple rule-based BE2 that does not take feasibility into account. The result of ULGR's advanced GR mechanism is its ability to efficiently generate columns with sufficiently large negative reduced costs to speed up convergence. To illustrate this, we provide the following statistics on the CG procedure in Table 3, the mean number of CG iterations and mean run-times for the PP for all three methods and values of $n$.

| $n$ | Avg. Nr. of CG iter. | | | Avg. time per iter (s) | | |
|---|---|---|---|---|---|---|
| | ULGR | BE2 | No GR | ULGR | BE2 | No GR |
| 200 | 289 | 182 | 172 | 10.71 | 15.14 | 19.13 |
| 500 | 491 | 325 | 359 | 6.65 | 10.20 | 9.22 |
| 1000 | 904 | 640 | 1212 | 3.62 | 5.16 | 2.81 |

**Table 3** Computational statistics for both GR techniques and no reduction on simulated data.

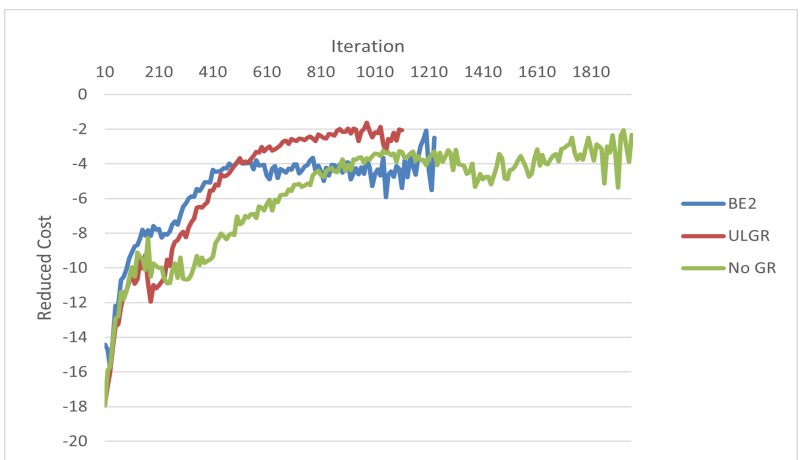

**Figure 4** Mean reduced costs averaged over the $K = 50$ instances against number of iterations for the three methods.

ULGR appears to have a larger number of CG iterations than BE2 for all cases. However, the run-times per CG iteration are always lower for ULGR, explaining why it is faster. A similar pattern holds when ULGR is compared to no reduction for the cases with $n = 200$ and 500. For $n = 1000$, the pattern differs as no reduction leads to more iterations with shorter run-times. Yet ULGR was still able to outperform no reduction for those instances as demonstrated by the last row of Table 2. This reinforces the verdict on ULGR's ability to reduce the PP while retaining its most valuable arcs.

Moreover, we plot the mean reduced cost over all $R = 50$ instances realized over the course of CG iterations for $n = 1000$ in Figure 4. With ULGR, the mean reduced costs converge to zero much faster. BE2, as expected, has a very strong start, as it focuses on arcs with minimal length. However, at later iterations where the PP is more difficult to solve, it suffers considerable delay, even compared to no reduction.

Our choice to apply the ULGR mechanism in a CG framework instead of solving C-VRPTW directly offers the potential to generate better integer solutions since CG generates lower bounds on the objective value of the master problem which can be used to produce better integer solutions rather than just the one solution from directly solving the master problem (Bredström et al., 2014). Additionally, having a reliable framework that offers an opportunity to efficiently solve a multitude of problems at large scale where the pricing problem is ESPPRC. Nonetheless, our framework can easily extend to other routing problems beyond ESPPRC as long as feasibility and contributions to the objective function and constraints can be indepednetly defined for each arc in the graph.

### 5.4 Model Generalization

In this section we study the performance of ULGR on the large-scale instances from Gehring & Homberger (1999). Since some of these instances have a different size than the instances on which our models are based, we populate the input to our UL models with 'dummy' customers who are always infeasible to reach ($q_{ij} = 2$ for dummy customer $j$ or $i$ or both, see Section 5.1). Thereafter, we remove the dummy customers from the output matrix $\mathcal{T}$ before calculating the heat map. For example, for $n = 400$, we use the model for instances of size 500 where 100 dummy customers are added to the original 400. The resulting $\mathcal{T}$ of length 500 is then reduced to dimensions 400 by eliminating all the rows and columns associated with the 100 dummy customers.

As a preliminary step, we further scale the coordinates and dual values that are provided as input to our GNN to be aligned with the ones observed during training. As such, we scale the coordinates by the largest coordinate value rounded up to the nearest 100 and the dual values by the depot's time window divided by 2.

The results are given in Tables 4 and 5 for both BE2 and no GR. ULGR outperforms BE2 for $n = 200$ as well as 800 and 1000. This is reflected in the negative objective gaps as well as the number of instances where $obj_{ULGR} < obj_{BE2}$. For $n = 800$ and 1000, we observe very small values for $t_{Speed-up}^{obj_{BE2} < obj_{ULGR}}$. This could be explained by the failure of our method to improve the starting objective value which happens in most of the instances where $BE2$ scores a better objective value. For $n = 600$, ULGR achieves BE2's objective value on average in around 77% of the latter's run-time for 45 of the 60 instances.

| $n$ | $obj_{Gap}$ | $J(<)$ | $t_{Speed-up}^{obj_{ULGR} < obj_{BE2}}$ | $t_{Speed-up}^{obj_{BE2} < obj_{ULGR}}$ |
|---|---|---|---|---|
| 200 | -0.99% | 37/60 | 0.58 | 0.72 |
| 400 | 1.47% | 29/60 | 0.79 | 0.75 |
| 600 | 1.04% | 45/60 | 0.77 | 0.47 |
| 800 | -4.52% | 50/60 | 0.33 | 0.07 |
| 1000 | -9.05% | 51/60 | 0.30 | 0.00 |

**Table 4** Results of ULGR technique compared to the DP baseline with BE2 reduction on the benchmark dataset.

Compared to the case when no reduction is employed, ULGR obtains lower objective values for most of the smaller instances of size 200 and 400. This is complemented by significant reductions in run-times as observed by the time-ratios. For the larger instances, ULGR delivers higher objective values for most instances while it does not seem to reduce run-times. However, the observed values of $t_{Speed-up}^{obj_{BE2} < obj_{ULGR}}$ are heavily discounted by some instances where both methods fail to find any improvement, yet the 'final' objective value of ULGR is reached by the baseline without reduction almost instantaneously.

| $n$ | $obj_{Gap}$ | $J(<))$ | $t_{Speed-up}^{obj_{ULGR} < obj_{NoGR}}$ | $t_{Speed-up}^{obj_{NoGR} < obj_{ULGR}}$ |
|---|---|---|---|---|
| 200 | -1.99% | 37/60 | 0.54 | 0.91 |
| 400 | -0.64% | 35/60 | 0.66 | 0.77 |
| 600 | 0.64% | 22/60 | 0.65 | 0.59 |
| 800 | 1.06% | 17/60 | 0.52 | 0.55 |
| 1000 | 1.05% | 15/60 | 0.54 | 0.48 |

**Table 5** Results of ULGR technique compared to the DP baseline without GR on the benchmark dataset.

Several main factors contribute to the decline in ULGR's performance. First and foremost is the differing distribution of input data. Secondly, the dummy customers used to populate input matrices may induce computational noise in the forward passage of the GNN. Our scaling of input to the GNN may also be a cause given its significant impact, as observed in our experiments. These issues may have a limited impact on the reduced search space for small instances of size close to 200. However, for larger instances, the performance gradually deteriorates, although ULGR can still be of use compared to BE2 for very large instances as it avoids premature termination.

The solution to dealing with instances from a different distribution is to retrain a new model. This is feasible given the short training times of ULGR, which are less than 3 hours even for the largest instances. Such a training time is very acceptable given that the model can be repeatedly deployed for any new input distribution, which does not change regularly or dramatically in practice.

# 6 Conclusions

In this paper, we present a method for solving large scale Combinatorial Optimization Problems with Column Generation where the Pricing Problem is an Elementary Shortest Path Problem with Resource Constraints (ESSPRC). The focus is on minimizing the objective values within a fixed computational budget. To that

end, we propose an Unsupervised Learning model that reduces ESPPRC so it can be solved efficiently by a local search heuristic.

We explain how the proposed framework could be extended to other problems rather than just Pricing Problems. Our model not only reduces the Pricing Problem significantly but also retains the most promising arcs, contributing to faster convergence. We benchmark our method against a reduction heuristic from the literature as well as to no reduction in the Pricing Problem for Capacitated Vehicle Routing Problem with Time Windows. For instances from a similar distribution, we achieved improvements in the objective value of up to 9.20% as well as reductions in run-time to reach the benchmark objective values of about 50%.

For instances from a different distribution, our model generalizes to smaller instances, but does not scale well with larger instances. We explain that the short training times conceived by our model enable easy reconfiguration to different distributions.

In presenting this study, we have established a reliable framework for Graph Reduction in Combinatorial Optimization. We plan to test our method on other Combinatorial Optimization problems in the future.

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
