# OpenReview forum: "Graph Reduction with Unsupervised Learning in Column Generation: A Routing Application"
_TMLR — Withdrawn by Authors_

### Review · Reviewer_oEhH · 2025-07-07

**Summary Of Contributions:**

This paper proposes a new heuristic to improve column generation for the Vehicle Routing Problem (VRP). In this setting, the pricing subroblem is formulated as an Elementary Shortest Path Problem with Resource Constraints (ESPPRC), which is hard to solve to optimality. The authors therefore propose adapting a graph reduction technique (ULGR) based on a learned heatmap, indicating which edges are likely to be part of a column with negative reduced cost. The underlying Graph Neural Network (GNN) is trained in an unsupervised fashion, using a loss that implicitly enforces cycle and resource constraints on the generated edge probabilities. Once the heatmap is generated, it forms the basis of a local search to find feasible paths that are added to the main problem. The method is evaluated against two learning-free column generation baselines on a set of random and realistic instances for the Capaciated VRP with Time Windows (CVRPTW).

**Audience:**

No

**Broader Impact Concerns:**

There are no unadressed ethical implications.

**Claims And Evidence:**

No

**Requested Changes:**

Here are the main changes that I deem necessary to improve the paper (see "strengths and weaknesses" for details):

**Claims and evidence**

- Include more competitive baselines in the experiments.
- Better justify algorithmic design choices, including through an ablation study.

**Audience**

- Introduce the mathematical details of the problem (CVRPTW, ESPPRC, column generation) to settle the framework and notations.
- Clarify the experimental section, its tables and figures, to better present algorithm comparisons.

---

I am currently leaning towards rejection, because the changes listed above seem too ambitious to be undertaken within a revision timeframe. I hope the authors do not take this as a hard no: the idea itself is interesting and worth pursuing, but my personal stance is that it needs more work before resubmission.

**Strengths And Weaknesses:**

## Strengths

**Claims and evidence**

- The idea of adapting heatmap-based unsupervised learning (originally proposed for the TSP) to the VRP's pricing problem is sound.
- Specific requirements of column generation for the VRP (resource constraints, presence of dual variables) are taken into account for the learning problem.

**Audience**

- Formulating unsupervised learning problems for combinatorial optimization is of great practical interest, because it dispenses users from the costly generation of a training set with optimal solutions.

## Weaknesses

**Claims and evidence**

- The experimental evidence is not sufficient to support the proposed method:
  - ULGR is only compared to column generation without graph reduction, or with one learning-free graph reduction baseline (BE2).
  - Previous learning-enhanced approaches to column generation, like [Morabit et al. (2023)](https://pubsonline.informs.org/doi/abs/10.1287/ijoo.2022.0082) or [Xu et al. (2023)](https://www.sciencedirect.com/science/article/pii/S0360835225002840) are not included for comparison, even though they also focus on the VRPTW, and even though their poor scaling is mentioned in the introduction.
  - Alternative models based on RL like [Kool et al. (2018)](https://openreview.net/forum?id=ByxBFsRqYm) are not benchmarked either, even though section 5.3 claims that they are less sample- and time-efficient during training.
  - Most of the experiments focus on randomly generated instances, instead of the most realistic instances recommended by e.g. [Accorsi et al. (2022)](https://www.sciencedirect.com/science/article/pii/S0167637722000244). The realistic instances are only used to study generalization. As for the random instances, they do not seem to follow any of the other established generation schemes (or if they do, it is not referred to).
  - Compared to the baseline, it is unclear what each component contributes to the performance, and no ablation study is performed. What is the respective impact of the neural architecture, the heatmap generation and adjustment, the loss function, the local search strategy? For instance, the claim on page 3 that "SAG is shown to be the most promising" is never backed by evidence.
  - The choice of hyperparameters is crucial for the performance of both ULGR (e.g. $M$) and BE2 (e.g. $\beta$) but no grid search is demonstrated to justify the values chosen. As a result, the graph reduction techniques output graphs of very different sizes, which (as admitted in section 5.3) makes comparison difficult.
  - On the largest instance in Table 3, the baseline without graph reduction actually boasts faster iterations, which seems to undermine the purpose of ULGR.
- The design choices underpinning the proposed method do not seem convincing enough:
  - The heatmap generation equation (1) was proposed by [Min et al. (2024)](https://proceedings.neurips.cc/paper_files/paper/2023/hash/93b8618a9061f8a55825c13ecf28392b-Abstract-Conference.html) in the context of the TSP. They justify it with their Corollary D.4, showing that in the ideal case where the edge selection probabilities in $\mathbb{T}$ are 0/1, then $\mathcal{H}$ represents a Hamiltonian cycle. It is unclear whether this is still the behavior we want for a routing subproblem, where the path does not have to go through every vertex in the graph.
  - Since training is performed independently from column generation, dual variables are sampled following the method of [Abouelrous et al. (2025)](https://arxiv.org/abs/2504.02383). The relevance of this sampling method for practical applications inside column generation is never questioned.
  - It is not explained why a GNN architecture, which is theoretically size-agnostic, does not scale to a different number of customers (which prompts the addition of dummy customers in section 5.4).
- Some previous works are misrepresented. For instance, in Section 3, it is not mentioned that the graph reduction heuristic by Desaulniers et al. (2008) does in fact account for connectivity, as stated by Xu et al. (2024).

**Audience**

- The problem is not presented or motivated adequately:
  - The paper does not contain a mathematical statement for the CVRPTW, or for the ESPPRC, or for the column generation algorithm. Notions like "dominant columns" are never introduced.
  - There is no explanation, even intuitive, as to why the ESPPRC is the right subproblem to focus on.
  - The content and handling of constraints is vague. For instance, in section 4.2, it is stated that feasibility of an arc is the first factor to consider in the heatmap, but feasibility is not defined anywhere.
- The overall exposition can be improved, in particular with respect to the experiments:
  - The specific features used for learning are never described in detail.
  - The three algorithms compared (ULGR, BE2 and simple column generation without graph reduction) are compared 2 by 2, instead of in a single table.
  - The split between both speed-up measures (depending on which algorithm got the best objective) is confusing and hard to parse. In addition, I think there might be an error or typo in equation (5), and in the subsequent sentences defining the other speed-up measure.
  - Standard deviations are not reported. For some instance sizes, bar charts are provided, but they are sometimes uninformative as well (e.g. figure 2c, where we only want to know whether the time ratio is > 1, and there is a single bar for the interval $[0, 2]$).
  - It is often hard to know from the captions which instances the corresponding figures refer to (in particular their size), or what the axes mean. For instance, in figure 2a, does the "time" correspond to training time or column generation iteration? why not represent the BE2 evolution too?
  - Figure 1 is very blurry.
  - The formatting of the experimental section leaves a lot of white space.

---

### Review · Reviewer_STHf · 2025-07-07

**Summary Of Contributions:**

The paper proposes an approach for unsupervised graph reduction for solving the pricing problem in column generation in vehicle routing problems. Specifically, the proposed approach extends a previous approach by Min et al. designed for TSPs to ESPPRC which represents the pricing problem in several classes of combinatorial optimization problems. The extension generalizes Min et al. to a more complex setting where not all nodes must be visited in a solution. Experiments on CVRPTW instances show the proposed approach outperform a heuristic graph reduction technique called BE2.

**Audience:**

Yes

**Broader Impact Concerns:**

No concerns.

**Claims And Evidence:**

No

**Requested Changes:**

Critical:
- Clarify framing and correspondingly justify the evaluation strategy
- Include evaluation of the solution quality of the GR
- Include comparison with existing baselines (e.g., Morabit et al. (2023) and Xu et al. (2023)).
- Clarify hyperparameter selection procedure

Changes that would strengthen the work:
- Include ablation study on the components of the approach
- Include additional problems (beyond CVRPTW)
- Include additional baselines (not necessarily CG-based) for CVRPTW
- Clarify whether the proposed approach guarantees feasibility and connectivity

**Strengths And Weaknesses:**

Strengths:
- Neural approaches for combinatorial optimization is an important research area that has received significant attention
- Experiments show that the approach outperforms the baseline (BE2)
- The paper is largely clear and well-written

----

Weaknesses:

**Framing is confusing:** it is really not clear whether the work is focused on how to solve the Elementary Shortest Path Problem with Resource Constraints (ESPPRC) problem, which is what the methodology is about, or how to solve the Capacitated Vehicle Routing Problems with Time Windows, which is what the evaluation seems to focus on. This discrepancy is confusing as we are not really evaluating the proposed methodology but a potential downstream application. Correspondingly it is not entirely clear whether the evaluation strategy, including the baselines and evaluation metrics are most appropriate ones to evaluate the proposed approach (more details in the following points).

**Limited experimental evaluation does is not sufficient for supporting the claims and for reproducibility:**
- Although the proposed approach focuses on solving the graph reduction problem for ESPPRC, if I understood correctly, the evaluation seems to focus on the solution quality for the downstream CVRPTW problem rather than the ESPPRC. There should be evaluation of the quality of solution for the GR problem (e.g., vs. the optimal solution or a strong baseline).
- No comparison to the existing supervised/RL baselines (Morabit et al. (2023) and Xu et al. (2023)).
- The various components of the approach are not evaluated in an ablation study: how will performance change if we drop each component in the new loss function? How will performance change if we utilize the approach by Min et al (designed for TSP) to generate the heat maps that are fed to the heuristic that solves the reduction problem?
- If the current evaluation is focused on the downstream CVRPTW, it is not clear why the evaluation is restricted to one baseline, and specifically only one that matches the current GR-based scheme for solving CVRPTW rather than other SOTA learning approaches for CVRPTW.
- The restriction to one downstream problem (CVRPTW) weakens the support for the claim that the approach is relevant to many other CO problems including other vehicle routing and crew scheduling problems.
- Reproducibility: how were the values of the hyperparameters (e.g., the weights \Lamba_1, ….) selected?

**Additional claim that needs further support:**
- "In doing so, we account for arc feasibility and graph-connectivity. This is in contrast to previous works such as Desaulniers et al. (2008), Dell’Amico et al. (2006) and Santini et al. (2018) where simple reduction heuristics have been applied to the PP, disregarding arc feasibility and connectivity.” - it is not clear if there is any support (theoretical or experimental) that the approach is indeed guarantees feasibility and connectivity?

**Limited technical contribution:** the technical contribution is somewhat limited and consists of an extension of Min et al. (2024) by generalizing it to a more complex setting where not all nodes must be visited in a solution. This seems to primarily involve the adaptation of the loss function and utilizing a different local search procedure following Guerriero et al. (2019).

**Minor typos:**
- “needed.Alternative” -> “needed. Alternative” (page 2)
- “term.The” -> “term. The” (page 5)

---

### Review · Reviewer_Tefj · 2025-07-08

**Summary Of Contributions:**

This paper introduces a new method extending prior work on learning to reduce graphs via column generation (SAG) to more complex vehicle routing problem. Similarly to SAG, the authors train a GNN via unsupervised learning to identify promising arcs to be kept, adapting components including the loss function, GNN input, heat map, and local search for the problem solved (CVRPTW). The proposed approach speeds up convergence compared to heuristics from the literature when applied to problems of similar structure to the ones seen during training.

**Audience:**

Yes

**Claims And Evidence:**

Yes

**Requested Changes:**

1. A comparison (ideally) or further discussion should be done for [1] and [2]. Why is your approach better, and how would it perform compared to those?

2. Explain why the heatmap should be symmetric.

3. What is the value of the penalty terms, and how were they set?

**Strengths And Weaknesses:**

### Strenghts

1. The paper is overall clearly written and suitable for the TMLR audience given its focus on the relevant CVRPTW.

2. Relevant literature is mentioned.

3. The method demonstrates strong performance against the BE2 heuristic.


### Weaknesses

1. My main concern is about a comparison against [1] and [2] which are mentioned in the paper but not compared against. Note that both [1] and [2] use ML methods to solve, among others, the CVRPTW. Moreover, only BE2 is compared for baseline heustic, while here are several other heuristics, as explained in [1].

2. It is unclear why we should symmetrize heatmap $\bar{\mathcal{H}}$ . In such a case, arcs going into the depot would have the same weight as the ones going out. No explanation is provided.

3. There are several proposed parts which are not studied. For instance, the loss function (2) has 3 penalty terms, but these are not provided, nor is an ablation study done to verify that this loss function actually provides better results.

4. It is hard to contextualize the results. No absolute value of the cost function is provided (e.g. gap to BKS or gap to HGS [3]), which makes it hard to compare against other papers.

5. As mentioned by the authors, generalization can be an issue for the proposed method, although I do not think this is too important.

6. (minor) There are a few issues with style, such as huge white space on pages 10 and 12. Moreover, figures are not vectorial or in PDF, which results in low quality (see e.g. the blur of Figure 1). I suggest including PDF versions in the manuscript instead.


[1] Xu, Kuan, Li Shen, and Lindong Liu. "Enhancing column generation by reinforcement learning-based hyper-heuristic for vehicle routing and scheduling problems." Computers & Industrial Engineering (2025): 111138.

[2] Morabit, Mouad, Guy Desaulniers, and Andrea Lodi. "Machine-learning–based arc selection for constrained shortest path problems in column generation." INFORMS Journal on Optimization 5, no. 2 (2023): 191-210.

[3] Vidal, Thibaut. "Hybrid genetic search for the CVRP: Open-source implementation and SWAP\* neighborhood." Computers & Operations Research 140 (2022): 105643.

---

### Note · Authors · 2025-07-23

**Comment:**

We thank all the reviewers for their valuable feedback. We understand the limitations of our study posed by the lack of availability of open-source implementations of proposed ML and CG baselines. Further investigations of algorithmic design require more extensive resources and time for which we are currently constrained. As such, we have taken the difficult decision of withdrawing this submission and improving it as much as possible in line with your guidance.

**Withdrawal Confirmation:**

I have read and agree with the venue's withdrawal policy on behalf of myself and my co-authors.